# Melatonin Supplementation Alters Maternal and Fetal Amino Acid Concentrations and Placental Nutrient Transporters in a Nutrient Restriction Bovine Model

**DOI:** 10.3390/metabo12121208

**Published:** 2022-12-02

**Authors:** Rebecca Swanson, Zully Contreras-Correa, Thu Dinh, Heath King, Darcie Sidelinger, Derris Burnett, Caleb Lemley

**Affiliations:** 1Department of Animal and Dairy Sciences, Mississippi State University, Starkville, MS 39762, USA; 2College of Veterinary Medicine, Mississippi State University, Starkville, MS 39762, USA

**Keywords:** nutrient restriction, melatonin, amino acids, placental nutrient transporters, intrauterine growth restriction, developmental programming

## Abstract

Melatonin rescues uterine blood flow and fetal body weight in a seasonal dependent manner within a nutrient restriction bovine model. We sought to identify the effects of nutrient restriction, melatonin, and sampling time on maternal and fetal amino acids, and placental nutrient transporters. Pregnant heifers received adequate or restricted nutrition, and 20 mg of melatonin or placebo from gestational days 160–240 over two seasons. On day 240 maternal and fetal blood, amnion, and placentomes were collected. Amino acid concentrations were determined by gas chromatography-mass spectrometry. Caruncle and cotyledon tissues were assessed for nutrient transporter density by qPCR. Data were analyzed using the MIXED procedure of SAS for fixed effects. In fall, melatonin rescued effects of nutrient restriction on System N, Anion, and total maternal amino acids. Furthermore, melatonin rescued effects of nutrient restriction on Systems A, N, Br, Bo, and essential amnion amino acids. In summer, melatonin rescued effects of nutrient restriction in Systems Br and Bo maternal amino acids. Furthermore, melatonin rescued effects of nutrient restriction on caruncle SLC38A10 and SLC38A2. Melatonin rescued effects of nutrient restriction in a seasonal dependent manner. These data align with previous studies suggesting melatonin is a more effective therapeutic in summer months.

## 1. Introduction

Compromised pregnancies are characterized by placental insufficiency resulting in poor fetal growth and development, and increased morbidity and mortality postnatally. Placental insufficiency is caused by decreased uteroplacental blood flow, nutrient transport, and placental size, resulting in intrauterine growth restricted (IUGR) offspring. Compromised pregnancies, including maternal under nutrition, are a concern in livestock production and human pregnancies. The rate of placental growth is less than that of fetal growth during the last third of gestation, thus, increased uterine blood flow is necessary to provide nutrients to the fetus [1]. Furthermore, nutrient transport across the placenta is imperative for proper growth, development, and survivability of the fetus [2]. Maternal nutrient restriction during gestation has been shown to decrease uterine blood flow and fetal body weight in sheep and cattle [3,4,5]. Furthermore, maternal nutrient restriction reduces amino acid transporter abundance in placental tissue of sheep and cows [6,7]. Intrauterine growth restriction, low birth weight, and poor postnatal growth reduce efficiency of livestock production and increase morbidity and mortality of children, thus, creating a need for therapeutics to improve placental insufficiency.

Melatonin is commonly known to regulate circadian rhythms in mammals and has antioxidant properties that can reduce reactive oxygen species often associated with compromised pregnancies [8]. Previous research has shown that melatonin increases antioxidant capacity in dairy cows [9]. Additionally, melatonin has vasodilative and vasoconstrictive properties that may regulate and redistribute blood flow [10]. Recently, dietary melatonin supplementation was shown to increase uterine blood flow and rescue fetal weight in fall calving heifers that were nutrient restricted over the summer months [4]. Interestingly, a seasonal discrepancy was observed in this study as melatonin supplementation failed to rescue uterine blood flow and fetal weight in nutrient restricted cows during the fall. Therefore, there is a need to investigate the therapeutic effects of melatonin supplementation during compromised pregnancies, specifically on amino acids in maternal, fetal, and amnion of a nutrient restriction bovine model.

Amino acids are the building blocks of protein, but they also have various roles in metabolic pathways that are vital for fetal growth and development throughout gestation [11]. Amino acids are necessary for placental and fetal growth and survivability, with fetal requirements increasing during the last third of pregnancy during the period of exponential growth [11]. Amino acid concentrations are greater in amnion and allantois fluids than in maternal blood in humans and livestock, indicating increased transport across the placenta [12]. In humans and in pig models of compromised pregnancies, amino acid transport across the placenta is decreased resulting in poor fetal growth [13,14]. In a maternal nutrient restriction sheep model, placentomes from IUGR pregnancies had decreased cationic and neutral amino acid transporters [6]. In an early gestation beef heifer nutrient restriction model, neutral and acidic amino acid transporters were reduced in placental tissue of restricted heifers [7]. Supplementation with amino acids that have angiogenic and vasodilative properties have improved uterine blood flow and fetal growth in humans, pigs, sheep, and mice [12]. Evaluation of amino acid transport systems in compromised pregnancies has been characterized in humans [14,15] sheep, and rats [13,16] while characterization of transport systems in bovines is limited [7]. Increased placental transporter density is necessary to facilitate increased amino acid demand during the last third of pregnancy, which is vital for fetal growth and development, as well as postnatal health and growth.

We hypothesized that nutrient restriction would reduce amino acid concentrations in maternal and fetal blood, and placental nutrient transporter transcript abundance, but melatonin would rescue these effects. Furthermore, we hypothesized increased amino acid concentrations and placental nutrient transporter transcript abundance post-feeding compared with pre-feeding sampling. Therefore, the objective of the current study was to evaluate the effects of nutrient restriction, melatonin supplementation, and time of sampling on maternal blood, fetal blood, and amnion amino acid concentrations by nutrient transport system and placental nutrient transporter transcript abundance.

## 2. Materials and Methods

### 2.1. Animals and Experimental Design

Animal management and treatments were performed at Mississippi State University’s H.H. Leveck Animal Research Center under Mississippi State University Institutional Animal Care and Use Committee protocol #17-709. Animal management was described previously [4]. Briefly, fifty-four Brangus crossbred heifers averaging 466 ± 39 kg were utilized in this study over a two-year period. Twenty-nine spring calving heifers and twenty-five fall calving heifers were used in the fall and summer replicates, respectively. Heifers were purchased from commercial breeders in the southeast United States. Heifers were bred via artificial insemination to a single sire and transitioned to a total mixed ration consisting of hay and base diet [4]. On day 140 of gestation, heifers were acclimated and trained to acquire feed from the Calan Broadbent feeding system (American Calan, Northwood, NH, USA). On day 160 of gestation heifers were stratified by body weight to one of four treatment groups: adequately fed (ADQ-CON) at 100% dietary needs, restricted (RES-CON) at 60% dietary needs, adequately fed with 20 mg of melatonin (AQD-MEL) or restricted with 20 mg of melatonin (RES-MEL) until 240 ± 2 days of gestation. Animal distribution across treatments was ADQ-CON (fall n = 7; summer n = 6), RES-CON (fall n = 7; summer n = 6), ADQ-MEL (fall n = 7; summer n = 6), and RES-MEL (fall n = 8; summer n = 7).

Control diets were fed at 2.4% BW/d with a target gain of 1.0 kg/d, while restricted heifers received 60% of the control diet [4]. All heifers were fed a grain mix used to top-dress treatment on feed daily [4]. In previous data from this project nutrient restriction significantly reduced heifer and fetal body weights in both seasons, while melatonin rescued fetal body weight in the summer [4]. Melatonin (Cayman Chemical Company, Ann Arbor, MI, USA; 20 mg) was dissolved in absolute ethanol and top-dressed on grain mix, while control groups received absolute ethanol as a placebo daily at 0900 h followed by diet delivery [4]. At day 240 ± 2 of gestation, Cesarean sections were performed on heifers at 0500 h (AM; fall n = 15, summer n = 12) or 1300 h (PM; fall n = 14, summer n = 13). Surgical procedures for fetal blood and placentome collection were according to Lemley, Hart, Lemire, King, Hopper, Park, Rude and Burnett [5].

### 2.2. Blood Parameters

Maternal blood was collected via coccygeal venipuncture into 6 mL EDTA vacutainer tubes (BD, Franklin Lakes, NJ, USA). Fetal blood was collected via trunk exsanguination into 6 mL EDTA vacutainer tubes. Amniotic fluid was collected during surgery via puncture of the amnion membrane with a sterile syringe. Plasma was isolated by centrifugation at 2000× *g* for 12 min at 4 °C and then stored at −80 °C until analysis. Amino acids from 100 µL of fluids were processed by using the EZfaast Amino Acid kit (Phenomenex^®^ Inc., Torrance, CA, USA), following the derivatization procedure developed by Kaspar et al. [17]. Briefly, the extracted amino acid solution was combined with an internal standard solution (0.02 nM of norvaline) and deproteinated through solid-phase extraction. The amino acids were then reacted with propyl chloroformate in chloroform, sodium hydroxide, and n-propanol. The derivatives were extracted in isooctane, evaporated, reconstituted in isooctane/chloroform mixture (4/1, *v*/*v*), and transferred to a 2-mL amber glass vial with a fixed insert (Agilent Technologies, Santa Clara, CA, USA) for GC-MS determination. Amino acid derivatives were injected into an inlet of an Agilent 7890A GC System coupled to an Agilent 5975C inert XL MSD with triple-axis mass detector, an Agilent 7693 Series Autosampler, and a capillary column (Zebron™ EZ-AAA 10 m × 0.25 mm; Phenomenex^®^, Santa Clara, CA, USA). The inlet was operated at 250 °C and 1:15 split ratio. The helium carrier gas was at a 1 mL/min constant flow rate. The temperature of the transfer line, ion source, and quadrupole were 310, 240, and 180 °C, respectively. The oven was programmed initially at 110 °C and ramped up to 320 °C within 11 min. The solvent delay was 1.30 min. The MS was operated in a SIM (selected ion monitoring) mode and target and qualitative ions were selected according to the manufacturer’s recommendation. Amino acids were identified by retention time and target and qualitative ions and quantified by an internal calibration method using authentic standards provided with the kits. The amino acid concentration was expressed as micromole per liter of fluid (µmol/L). Propyl chloroformate does not derivatize arginine and allo-isoleucine and α-aminoadipic were below the quantification level in maternal plasma, fetal plasma, and amnion. Total and essential amino acid concentrations (µmol/L) were calculated by summation in maternal and fetal plasma and amnion. Furthermore, amino acids by transport system were summed in maternal and fetal plasma and amnion. System A amino acids includes the three favored neutral amino acids by SLC38A2 nutrient transporter. System Bo amino acids are neutral amino acids, excluding those is System A, and branched chain amino acids that can be transported by SLC7A5 and SLC7A8. System Br amino acids includes branched chain amino acids, which can work through various transporters. System N includes amino acids that use sodium-dependent transport, favoring SLC38A10. The amino acids included in each system and essential amino acids can be found in Appendix A.

### 2.3. Placentome Collection and Separation

After removal of the fetus, placentomes nearest the umbilicus were excised and separated into caruncle (maternal) and cotyledon (fetal) portions and snap frozen in liquid nitrogen and stored at −80 °C until further processing.

### 2.4. Gene Expression

Transcript abundance of nutrient transporters was determined in carunclular and cotyledonary tissue by RT-PCR. Approximately 30 mg of frozen tissue was used to isolate RNA using a miRNeasy Mini Kit (QIAGEN, Hilden, Germany) according to manufacturer protocol. Isolated RNA was calculated using a NanoDrop One spectrophotometer (Thermo Scientific, Waltham, WA, USA) and frozen at −80 °C until further processing. Synthesis of cDNA was done using High-Capacity cDNA Reverse Transcription Kit (Thermo Fisher Scientific, Auburn, AL, USA) following manufacturer protocol. Custom Taqman Gene Expression Assays (Thermo Fisher Scientific, Auburn, AL, USA) were used in duplex to perform RT-PCR according to manufacturer protocol using QuantStudio 3 (Applied Biosystems, Foster City, CA, USA) for real-time qPCR. The genes of interest were solute carrier family 2 member 8 (SLC2A8), solute carrier family 43 member 2 (SLC43A2), solute carrier family 7 member 5 (SLC7A5), solute carrier family 7 member 8 (SLC7A8), solute carrier family 38 member 10 (SLC38A10), and solute carrier family 38 member 2 (SLC38A2). Ribosomal protein L19 (RPL19) and splicing factor 3a subunit 1 (SF3A1) were used as housekeeping genes. Assays for the genes can be found in Table 1. Replicate CT values were averaged and used for relative quantification using the 2^−ΔΔCt^ method where the geometric mean of RPL19 and SF3A1 was used as a reference to normalize genes of interest.

### 2.5. Statistical Analysis

Normality was tested using the Shapiro–Wilks test of the UNIVARIATE procedure in SAS 9.4 (SAS Institute, Cary, NC, USA). Nonparametric data were ranked using the RANK procedure of SAS before analysis. All data were analyzed using the MIXED procedure of SAS to determine the effects of nutritional plane, melatonin supplement, and their interaction in a 2 × 2 factorial design. Furthermore, a main effect of sampling time was evaluated. Maternal initial body weight, maternal body weight at Cesarean section, day of gestational age, fetal sex, and fetal body weight were included as covariates in the model if *p* ≤ 0.20. Replicate was *p* < 0.10 for several parameters, thus, data sets were analyzed and presented separately for the fall and summer. Heifer was considered the experimental unit. All data are presented as means ± standard error. The threshold for significance was *p* ≤ 0.05.

## 3. Results

### 3.1. Plasma Amino Acids

Individual amino acids in maternal plasma of the fall trial are provided in Appendix A. In the fall trial, there were nutrition by treatment interactions (*p* < 0.05) in System N, Anion, and total amino acids in maternal plasma, whereby each classification was increased in RES-CON compared with ADQ-CON while RES-MEL was not different from ADQ-CON (Table 2). There was a nutrition by treatment interaction (*p* < 0.01) in System A amino acids, whereby RES-CON was increased compared to ADQ-CON and RES-MEL and RES-MEL was increased compared to ADQ-CON (Table 2). Furthermore, there was an effect (*p* < 0.05) of sex in System A amino acids in maternal plasma, which were decreased in males vs. females (data not shown). There was a nutrition by treatment interaction (*p* = 0.01) in Cationic amino acids, whereby RES-MEL was decreased compared to ADQ-MEL while ADQ-CON and RES-CON were intermediary (Table 2). Furthermore, System A amino acids in maternal plasma were reduced (*p* < 0.01) in the afternoon (602.83 µmol/L ± 95.72 µmol/L) compared with the morning (756.30 µmol/L ± 86.67 µmol/L). Furthermore, System Cationic amino acids were reduced (*p* < 0.01) in maternal plasma in the afternoon (105.07 µmol/L ± 6.86 µmol/L) compared to the morning (134.84 µmol/L ± 7.05 µmol/L). There were no differences among treatment groups, or sampling time in System Br, System Bo, or essential amino acids in maternal plasma.

Individual amino acids in fetal plasma in the fall trial are provided in Appendix A. In the fall trial, there were no differences among treatment groups, or sampling time in System A, System Br, System N, Cationic, total, and essential amino acids in fetal plasma. System Bo amino acids in fetal plasma were reduced (*p* = 0.01) among nutrient restricted compared with adequately fed heifers (Table 2). System Anion amino acids in fetal plasma were increased (*p* = 0.02) among MEL heifers compared with CON heifers (Table 2).

Individual amino acids in amnion in the fall trial are provided in Appendix A. In the fall trial, there were nutrition by treatment interactions (*p* < 0.05) in System A, System N, and total amino acids in amnion, whereby, RES-MEL and ADQ-MEL were increased compared with ADQ-CON while RES-MEL was intermediary (Table 2). There were nutrition by treatment interactions (*p* < 0.05) in System Br and System Bo amino acids in amnion, whereby, RES-CON was increased compared with ADQ-CON while RES-MEL was not different from ADQ-CON (Table 2). There was a nutrition by treatment interaction (*p* = 0.03) in Cationic amino acids, however, there were no differences in means separation (Table 2). There was a nutrition by treatment interaction (*p* = 0.01) in essential amino acids, whereby, RES-CON was increased compared to ADQ-CON while RES-MEL was not different from ADQ-CON (Table 2).

Individual amino acids in maternal plasma of the summer trial are provided in Appendix A. There were nutrition by treatment interactions (*p* < 0.05) in System Br and System Bo amino acids, whereby, RES-CON was decreased compared with RES-MEL while ADQ-CON and ADQ-MEL were intermediary (Table 3). Furthermore, System Br amino acids were decreased (*p* = 0.04) in the afternoon (562.56 µmol/L ± 20.37 µmol/L) compared with the morning (664.50 µmol/L ± 103.87 µmol/L). In the summer trial, there were no differences among treatment groups, or sampling time in System A, System N, Cationic, Anion, and total amino acids in maternal plasma.

Individual amino acids in fetal plasma during the summer trial are provided in Appendix A. In the summer trial, there were no differences among treatment groups, or sampling time of System A, System Bo, System N, and Cationic amino acids in fetal plasma. System Br amino acids in fetal plasma were decreased (*p* = 0.01) among nutrient restricted heifers compared with adequately fed heifers (Table 3). System Anion amino acids in fetal plasma were increased (*p* = 0.03) among melatonin supplemented versus control heifers (Table 3). Furthermore, there was an effect (*p* < 0.05) of sex in Anion amino acids in fetal plasma, which were increased in male vs. female (data not shown). Essential amino acids in fetal plasma were decreased (*p* = 0.02) in nutrient restricted heifers compared to adequately fed heifers (Table 3).

Individual amino acids in amnion of the summer trial are provided in Appendix A. In the summer trial, there were no differences among treatment groups or sampling time of System A, System Br, System Bo, System N, Cationic, total, and essential amino acids in amnion. There were effects (*p* < 0.05) of sex in System Br and essential amino acids in amnion, which were decreased in male vs. female (data not shown). Anion amino acids were decreased (*p* = 0.02) in nutrient restricted heifers compared with adequately fed heifers (Table 3).

### 3.2. Placenta Transcript Abundance

In the fall trial, caruncle SLC2A8 transcript abundance was decreased (*p* = 0.01) in the morning (0.9204 ± 0.2193 µmol/L) compared with the afternoon (1.3320 ± 0.2729 µmol/L). Caruncle SLC7A5 was increased (*p* = 0.05) among nutrient restricted heifers compared with adequately fed heifers (Table 4). Caruncle SLC38A10 was increased (*p* = 0.03) among nutrient restricted heifers vs. adequately fed heifers (Table 4). Caruncle SLC7A8 was increased (*p* = 0.04) in the morning (0.5905 ± 0.0850 µmol/L) compared with the afternoon (0.4261 ± 0.0478 µmol/L). There were no differences among treatment groups, or sampling time of caruncle SLC43A2 or SLC38A2.

In the fall trial, cotyledon SLC2A8 transcript abundance was increased (*p* = 0.03) in nutrient restricted vs. adequately fed heifers (Table 4). Cotyledon SLC7A8 was increased (*p* = 0.02) in melatonin supplemented heifers compared with control heifers (Table 4). Furthermore, there was an effect (*p* < 0.05) of sex in cotyledon SLC7A8, which was increased in males vs. females (data not shown).

In the summer trial, there were no differences among treatment groups or sampling time of caruncle SLC43A2, SLC7A5, or SLC7A8. There were nutrition by treatment interactions (*p* < 0.05) in caruncle SLC38A10 and SLC38A2, whereby, RES-CON was increased compared with ADQ-CON and RES-MEL was not different from ADQ-CON (Table 5). Furthermore, caruncle SLC38A2 was decreased (*p* < 0.01) in the morning (0.4259 ± 0.0628 µmol/L) compared with the afternoon (0.5887 ± 0.0831 µmol/L). Caruncle SLC2A8 was increased (*p* = 0.03) in the morning (1.0601 ± 0.1704 µmol/L) compared with the afternoon (0.7087 ± 0.0795 µmol/L).

In the summer trial, there were no differences among treatment groups, or sampling time in cotyledon SLC2A8, SLC43A2, SLC7A5, SLC7A8, SLC38A10, or SLC38A2.

## 4. Discussion

In this study, we found an abundance of differences in the fall replicate while differences were lesser in the summer replicate (Figure 1). Interestingly, this aligns with previous data from this project, in which uterine blood flow and fetal body weights were rescued in the melatonin supplemented nutrient restricted heifers in the summer replicate but not the fall replicate [4]. Therefore, limited differences in plasma amino acids in the summer replicate may be directly related to rescued uterine blood flow resulting in improved placental efficiency and normal amino acid delivery during melatonin treatment. However, we do recognize a limitation of this study is that uterine blood flow was not collected on the same day as fluids and placentomes. Furthermore, umbilical blood flow was not measured to identify flux across the placenta at sample collection due to time constraints to meet the second objective of this study. Seasonality appears to affect female reproduction in several ways, including cyclicity, conception, uterine blood flow, and offspring performance, which may be due to photoperiod, weather, or a combination of these environmental factors. In a study by Cain et al. [18], spring calving beef heifers had increased uterine blood flow during gestation and heavier calves at birth compared to fall calving heifers. Conversely, in a study by King and Macleod [19], postpartum fall calving cows returned to cyclicity and became pregnant earlier than spring calving cows. Similarly, in a study by Caldwell et al. [20], calving rates were higher and calving interval was shorter in fall calving cows compared to spring calving cows. Furthermore, fall born calves had increased weaning weights and average daily gain compared to spring born calves that were heaver at birth [20]. Therefore, melatonin may be a more effective therapeutic in certain seasons, which can direct future application of melatonin for optimal economic efficiency among livestock producers.

In maternal plasma of the fall replicate, System N, Cation, Anion, and total amino acid concentrations among nutrient restricted heifers were rescued by melatonin supplementation. Furthermore, System A amino acids among nutrient restricted heifers were partially improved by melatonin. While research evaluating amino acids in compromised pregnancies by transport systems has not previously been characterized in beef cows, these systems include all classifications of amino acids, including sodium-dependent, sodium-independent, cationic, neutral, basic, small, and large amino acids. In a sheep study by Lemley et al. [21], melatonin did not mitigate the effects of nutrient restriction on total or maternal branch chain amino acids, however melatonin did partially rescue uteroplacental total and branch chain amino acid flux. It is important to note that ewes sampled in this study were 18 h post-feeding, which is considerably different from sampling times in the present heifer study. In another sheep study by Chung et al. [22], maternal concentrations of amino acids were decreased compared to fetal concentrations, demonstrating the importance of efficient amino acid transport across the placenta. Notably, there were minimal differences in the summer, again showing the rescue effects of melatonin on uterine blood flow and fetal body weight diminishes the need for alterations to provide sufficient fetal nutrient delivery.

In fetal plasma of the fall replicate, System Bo amino acids were reduced among nutrient restricted heifers, while the only effect of melatonin was an increase in System Anion amino acids. In fetal plasma of the summer replicate, System Br and essential amino acids were decreased among nutrient restricted heifers. Furthermore, the fall replicate experienced greater fetal growth restriction, which was not rescued by dietary melatonin supplementation. These data are similar to a sheep study by Edwards, McKnight, Askelson, McKnight, Dunlap and Satterfield [6], in which most of the branch chain and essential amino acids were decreased in IUGR pregnancies. It has been demonstrated that fetal catabolism and subsequent uteroplacental uptake of essential and branch chain amino acids are important in proper placental function due to their role in metabolism and steroidogenesis [22]. Acute infusion of essential, branched-chain, and non-essential amino acids increased fetal amino acid concentrations, reduced protein breakdown, and increased protein accretion in IUGR sheep [23]. Similarly, prolonged infusion of essential, branched-chain and non-essential amino acids increased fetal uptake, fetal concentrations, and uteroplacental flux of branch chain and essential amino acids, as well as increased leucine oxidation in IUGR sheep [24]. Furthermore, chronic arginine infusion rescued lamb birth weights from nutrient restricted sheep [25]. These rescue effects of branch chain and essential amino acids in compromised pregnancies, including improved metabolic function in the fetus and placenta, as well as birth weights, reaffirm the importance of efficient nutrient transport for proper fetal growth. Taken together with our data set, showing seasonally dependent alterations in maternal and fetal amino acid concentrations, the therapeutic benefits of amino acid or protein supplementation during compromised pregnancies may have a greater efficacy during specific seasons.

In amnion of the fall replicate, System A, System Br, System Bo, System N, and essential amino acids were increased by maternal nutrient restriction and rescued by melatonin. Previously, bovine amnion amino acid concentrations influenced by melatonin have not been reported, however, a sheep study by Kwon et al. [26] showed reduced amino acids in amnion of mid- to late-gestation nutrient restricted ewes. Interestingly, previous sheep research has not reported seasonal alterations in amino acid concentrations as these studies occur during specific seasons or indoors with environmental control. Conversely, there were minimal effects of nutrition or treatment observed in amino acids of the summer replicate. Therefore, examining amino acid profiles in polyestrous species under changing environmental conditions during the 3^rd^ trimester of pregnancy could influence therapeutic supplementation strategies to mitigate compromised pregnancies and negative fetal programming responses. Amnion amino acid concentrations throughout gestation have been characterized in normal human pregnancies, which exhibit a decrease in amino acids in the 3rd trimester compared to the 2nd trimester [27]. This decrease in amnion amino acid concentrations during late gestation has been attributed to increased fetal utilization during final maturation and protein deposition [27]. In sheep IUGR models, amino acid supplementation has shown promise in rescuing the effects of hyperthermia-induced placental insufficiency, however, women consuming high protein diets to reverse the effects of IUGR had adverse outcomes on perinatal health which may be attributed to a lack of amino acid availability to the fetus [28]. Amino acid supplementation directly to amnion has been considered but requires invasive techniques. The present study suggests melatonin supplementation can alter amnion amino acid profiles in compromised pregnancies, and thus, serves as a model for less invasive methods to study therapeutics for IUGR pregnancies.

In caruncular tissue of both replicates, SLC38A10 was increased among nutrient restricted heifers, however, melatonin only rescued this effect in the summer replicate. Furthermore, SLC38A2 was also rescued by melatonin in nutrient restricted heifers from the summer replicate. Prior to the present study, the effects of melatonin on placental nutrient transporter expression in a compromised pregnancy has not been reported, however, in a sheep study by Edwards, McKnight, Askelson, McKnight, Dunlap and Satterfield [6], SLC38A2 abundance was reduced in placentomes from nutrient restricted IUGR pregnancies, which is contradictory to the current study. Previous work from our lab found genes associated with peptide biosynthesis and peptide metabolism were upregulated in placentomes from nutrient restricted heifers [29]. In early gestation nutrient restricted heifers, neutral amino acid transporters were reduced in uteroplacental tissues [7]. These studies contradict the present study, in which nutrient transporters are increased in caruncular tissue from nutrient restricted bovine, however, it is important the note the aforementioned studies combined uteroplacental tissue rather than separating maternal and fetal tissues alike the present study. There is a need for future studies to evaluate caruncle and cotyledon nutrient transporters in bovine compromised pregnancy models.

Interestingly, there were minimal effects of time, however there were more time effects in the fall replicate compared to the summer replicate. In the fall replicate, maternal System A, Cationic, and total amino acids were decreased in the afternoon sampling, which was 4 h post-feeding. While studies on alterations of circulating amino acids relative to feeding time are limited, a study by Forde et al. [30] found cationic amino acids and cationic transporters in the endometrium increase by day during early gestation. These findings suggest sampling time is an important consideration when planning data collection as the time may impact overall findings in a study. Furthermore, sample times may differ due to season, which might be attributed to day length and circadian rhythm.

Overall, this study implies melatonin is an effective therapeutic for amino acid and nutrient transporter alterations caused by nutrient restriction, however, more so during the summer months. Livestock models of developmental programming lack consideration of season and the respective environmental changes, which has been proven to impact amino acids and fetal growth in bovine, which are not seasonal breeders. The fall season may be more detrimental to placental efficiency compared to the summer season, suggesting a need for research into the seasonal effects of bovine pregnancy, including therapeutics, and management strategies. Specifically, future studies can evaluate seasonal differences in fetal metabolite profiles, postnatal growth, and health and well-being. Understanding these seasonal effects will aid in the use of therapeutics and the development of management techniques to ensure normal, healthy pregnancies, resulting in efficient postnatal growth and improved health status. While melatonin has been proven to be an effective therapeutic in the present study, melatonin studies should be replicated across models of IUGR, including various species and the method used to induce placental insufficiency (i.e., overnutrition, undernutrition, hyperthermia, multiples), and in various seasons or regions to understand the optimal use as a therapeutic. Furthermore, investigation into maternal, fetal, and amnion amino acid concentrations and placental nutrient transporters should be done at several time points, including both pre- and post-feeding.

## Figures and Tables

**Figure 1 metabolites-12-01208-f001:**
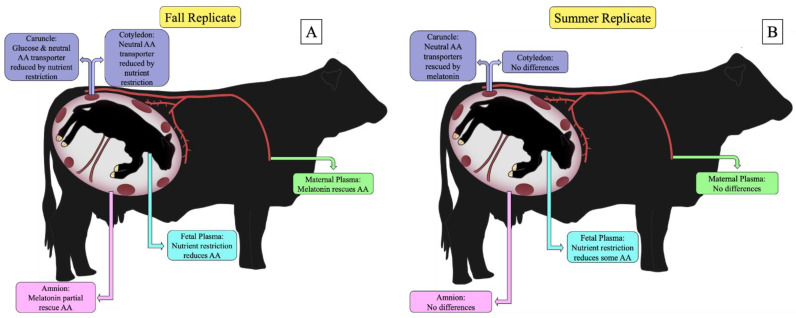
Summary of main findings in the fall and summer replicate of melatonin supplemented nutrient restricted cows. Including maternal, fetal, and amnion amino acid concentrations, and placental nutrient transporters in the caruncle and cotyledon. (**A**) Summary of findings in maternal and fetal blood, amnion, and placenta in the fall replicate; (**B**) Summary of findings in maternal and fetal blood, amnion, and placenta in the summary replicate.

**Table 1 metabolites-12-01208-t001:** Function, assays, accession, and amplicon length of primers used for TaqMan real-time PCR quantitation of placental gene expression. RPL19 and SF3A1 were used as housekeeping genes.

Gene	Function	Assay ID	Accession	Amplicon	
SLC2A8	Glucose and fructose transport	Bt03217733_m1	NP_963286	80	
SLC43A2	Sodium independent amino acid transport	Bt03246452_m1	NP_001069014.1	94	
SLC7A5	Large neutral amino acid transport	Bt03215392_m1	NP_777038.1	64	
SLC7A8	Large neutral amino acid transport	Bt04312788_m1	NP_001179818.1	66	
SLC38A10	Sodium dependent amino acid transport	Bt03267849_m1	BC153282	56	
SLC38A2	System A amino acid transport	Bt03255119_m1	NP_001075893.1	67	
RPL19	Reference gene	Bt03229687_g1	NP_001035606.1	82	
SF3A1	Reference gene	Bt03254301_m1	NP_001074979.1	60	

**Table 2 metabolites-12-01208-t002:** Amino acids in maternal and fetal plasma, and amnion by transport system, total amino acids, and essential amino acids in the fall replicate.

Fall Trial					*p*-Value		
Maternal Plasma	ADQ-CON	RES-CON	ADQ-MEL	RES-MEL	NUT	TRT	NUT*TRT
System A	512.03 ± 28.96 ^c^	1040.38 ± 166.26 ^a^	685.55 ± 122.33 ^b^	514.80 ± 68.89 ^b^	-	-	** <0.01 **
System Br	431.58 ± 32.82	383.23 ± 37.62	461.19 ± 34.67	461.59 ± 30.56	0.62	0.07	0.38
System Bo	540.01 ± 35.26	44.88 ± 41.08	580.98 ± 37.86	592.64 ± 32.98	0.87	0.16	0.91
System N	342.97 ± 37.54 ^b^	710.26 ± 115.26 ^a^	468.48 ± 81.27 ^ab^	338.95 ± 66.97 ^b^	-	-	** <0.01 **
Cationic	107.41 ± 9.66 ^ab^	132.37 ± 9.82 ^ab^	133.72 ± 9.93 ^a^	106.32 ± 8.97 ^b^	-	-	** 0.01 **
Anion	76.72 ± 2.58 ^c^	244.38 ± 63.22 ^a^	107.68 ± 17.07 ^ab^	90.01 ± 32.37 ^bc^	-	-	** <0.01 **
Total	1883.94 ± 49.14 ^b^	2676.84 ± 271.37 ^a^	2165.16 ± 189.28 ^ab^	1897.11 ± 134.93 ^b^	-	-	** 0.01 **
Essential	649.61 ± 32.86	620.72 ± 32.86	715.21 ± 32.86	661.74 ± 30.66	0.21	0.11	0.71
** Fetal Plasma **							
System A	1229.16 ± 137.88	1206.54 ± 137.88	1348.55 ± 137.88	1235.97 ± 128.66	0.62	0.59	0.74
System Br	363.87 ± 29.54	352.72 ± 30.34	344.12 ± 31.57	348.76 ± 27.94	0.92	0.69	0.79
System Bo	43.90 ± 54.90	510.92 ± 63.97	536.58 ± 58.95	518.16 ± 51.36	** 0.01 **	0.60	0.88
System N	770.52 ± 127.05	752.38 ± 127.05	883.46 ± 127.05	792.39 ± 118.55	0.67	0.55	0.77
Cationic	135.04 ± 13.56	130.53 ± 13.39	148.40 ± 13.39	129.37 ± 12.67	0.38	0.66	0.59
Anion	334.13 ± 43.75	256.46 ± 43.75	426.94 ± 43.75	385.70 ± 40.83	0.18	** 0.02 **	0.68
Total	3058.75 ± 207.49	2835.00 ± 207.49	3293.79 ± 207.49	3016.63 ± 193.61	0.23	0.32	0.90
Essential	634.49 ± 41.05	599.99 ± 42.17	615.92 ± 43.88	590.72 ± 38.83	0.50	0.74	0.91
** Amnion **							
System A	145.84 ± 26.37 ^b^	364.57 ± 67.50 ^a^	481.58 ± 180.53 ^a^	258.89 ± 108.28 ^ab^	-	-	** 0.01 **
System Br	34.43 ± 6.89 ^b^	256.31 ± 104.84 ^a^	215.38 ± 86.18 ^a^	38.25 ± 8.06 ^b^	-	-	** <0.01 **
System Bo	67.16 ± 9.66 ^b^	338.88 ± 125.65 ^a^	301.57 ± 109.36 ^ab^	74.85 ± 11.96 ^b^	-	-	** 0.01 **
System N	92.84 ± 12.88 ^b^	222.94 ± 49.27 ^ab^	290.22 ± 104.09 ^a^	145.94 ± 48.72 ^ab^	-	-	** 0.02 **
Cationic	42.51 ± 3.70	89.64 ± 20.56	89.97 ± 21.96	48.66 ± 7.65	-	-	** 0.03 **
Anion	42.64 ± 6.03	63.34 ± 10.56	54.22 ± 7.59	46.60 ± 7.08	0.12	0.38	0.10
Total	525.96 ± 74.69 ^b^	1296.53 ± 279.26 ^ab^	1409.06 ± 386.44 ^a^	744.10 ± 230.51 ^ab^	-	-	** 0.02 **
Essential	104.61 ± 13.72 ^b^	416.74 ± 144.12 ^a^	378.91 ± 126.68 ^a^	117.58 ± 23.33 ^b^	-	-	** 0.01 **

Data are presented as means ± standard error. Bolded *p* values are significant at *p* ≤ 0.05. When there is a significant interaction, the *p* values for main effects of nutrition and treatment are not shown and replaced with a dash (-). Superscripts (a, b) denotes differences at *p* ≤ 0.05. Concentrations are expressed in µmol/L. Experimental units: ADQ-CON, n = 7; RES-CON, n = 7; ADQ-MEL, n = 7; RES-MEL n = 8. Abbreviations: NUT, nutrition; TRT, treatment; NUT*TRT, nutrition by treatment interaction; ADQ-CON, adequately fed control; RES-CON, restricted fed control; ADQ-MEL, adequately fed melatonin supplemented; RES-MEL, restricted fed melatonin supplemented.

**Table 3 metabolites-12-01208-t003:** Amino acids in maternal and fetal plasma, and amnion by transport system, total amino acids, and essential amino acids in the summer replicate.

Summer Trial					*p* Value		
Maternal Plasma	ADQ-CON	RES-CON	ADQ-MEL	RES-MEL	NUT	TRT	NUT*TRT
System A	1020.97 ± 229.93	1076.96 ± 229.93	962.94 ± 229.93	859.07 ± 213.48	0.92	0.55	0.83
System Br	436.57 ± 29.94 ^ab^	376.50 ± 26.88 ^b^	412.14 ± 33.13 ^ab^	496.68 ± 27.46 ^a^	-	-	** 0.02 **
System Bo	596.56 ± 28.70 ^ab^	526.00 ± 28.62 ^b^	578.95 ± 34.49 ^ab^	645.74 ± 29.14 ^a^	-	-	** 0.03 **
System N	781.29 ± 148.09	724.37 ± 148.12	693.30 ± 148.04	637.47 ± 135.69	0.70	0.55	0.99
Cationic	107.23 ± 7.75	110.77 ± 9.80	133.75 ± 10.02	117.85 ± 10.39	0.59	0.09	0.45
Anion	116.99 ± 27.26	146.28 ± 44.54	179.18 ± 52.34	96.39 ± 25.46	0.77	0.89	0.37
Total	2481.94 ± 321.02	2525.55 ± 333.38	2753.47 ± 381.67	2146.38 ± 109.50	0.70	0.86	0.66
Essential	780.96 ± 38.02	625.66 ± 36.86	767.78 ± 48.44	729.11 ± 47.84	0.76	0.11	0.07
** Fetal Plasma **							
System A	1418.83 ± 111.75	1387.76 ± 264.42	1440.70 ± 113.34	1627.40 ± 55.52	0.21	0.56	0.53
System Br	397.96 ± 25.89	296.81 ± 51.25	424.59 ± 26.73	367.26 ± 14.05	** 0.01 **	0.19	0.90
System Bo	591.95 ± 31.54	473.28 ± 78.81	634.72 ± 37.87	544.50 ± 20.38	0.15	0.77	0.59
System N	992.98 ± 123.99	957.46 ± 123.96	885.32 ± 128.04	1160.96 ± 117.25	0.14	0.47	0.06
Cationic	77.54 ± 10.43	61.12 ± 6.96	81.90 ± 9.43	63.99 ± 7.37	0.64	0.71	0.75
Anion	142.34 ± 19.36	129.18 ± 20.48	214.14 ± 20.46	196.44 ± 19.13	0.58	** 0.03 **	0.67
Total	2298.56 ± 121.57	2122.10 ± 354.06	2397.82 ± 133.17	2479.26 ± 78.33	0.23	** 0.03 **	0.20
Essential	687.99 ± 38.02	552.43 ± 87.25	712.19 ± 37.09	616.02 ± 12.43	** 0.02 **	0.25	0.83
** Amnion **							
System A	524.93 ± 207.20	405.86 ± 60.36	355.05 ± 70.99	346.49 ± 89.86	0.80	0.22	0.84
System Br	331.21 ± 117.82	324.11 ± 95.51	196.19 ± 69.17	184.90 ± 91.63	0.50	0.09	0.72
System Bo	442.86 ± 146.00	420.16 ± 114.10	265.26 ± 83.17	247.35 ± 110.75	0.88	0.12	0.91
System N	294.15 ± 62.41	271.20 ± 52.98	218.59 ± 58.60	192.71 ± 55.30	0.67	0.19	0.98
Cationic	88.94 ± 14.20	85.73 ± 14.82	69.12 ± 10.52	93.47 ± 37.58	0.69	0.44	0.90
Anion	75.42 ± 15.42	49.62 ± 6.06	46.43 ± 5.07	45.01 ± 5.28	** 0.02 **	0.34	0.79
Total	1663.68 ± 341.91	1464.54 ± 290.30	1241.54 ± 314.54	1105.49 ± 299.78	0.60	0.22	0.92
Essential	517.51 ± 155.41	499.11 ± 127.43	340.53 ± 91.07	328.29 ± 116.51	0.38	0.18	0.74

Data are presented as means ± standard error. Bolded *p* values are significant at *p* ≤ 0.05. When there is a significant interaction, the *p* values for main effects of nutrition and treatment are not shown and replaced with a dash (-). Superscripts (a, b) denotes differences at *p* ≤ 0.05. Concentrations are expressed in µmol/L. Experimental units: ADQ-CON, n = 6; RES-CON, n = 6; ADQ-MEL, n = 6; RES-MEL n = 7. Abbreviations: NUT, nutrition; TRT, treatment; NUT*TRT, nutrition by treatment interaction; ADQ-CON, adequately fed control; RES-CON, restricted fed control; ADQ-MEL, adequately fed melatonin supplemented; RES-MEL, restricted fed melatonin supplemented.

**Table 4 metabolites-12-01208-t004:** Transcript abundance of nutrient transporters in caruncle and cotyledon tissue in the fall replicate.

Fall Trial					*p* Value		
Caruncle	ADQ-CON	RES-CON	ADQ-MEL	RES-MEL	NUT	TRT	NUT*TRT
SLC2A8	0.6893 ± 0.0617	1.8826 ± 0.5294	0.8714 ± 0.2051	1.0438 ± 0.3087	0.66	0.31	0.47
SLC43A2	0.7092 ± 0.1151	0.9292 ± 0.2260	0.8478 ± 0.1176	1.8979 ± 0.3620	0.20	0.11	0.25
SLC7A5	0.7013 ± 0.0774	1.7454 ± 0.5591	0.9267 ± 0.1778	1.2461 ± 0.2370	** 0.05 **	0.45	0.36
SLC7A8	0.5305 ± 0.0709	0.5901 ± 0.1676	0.4050 ± 0.0725	0.5180 ± 0.0835	0.93	0.30	0.43
SLC38A10	0.8010 ± 0.0760	1.0314 ± 0.1567	0.7262 ± 0.1105	1.1865 ± 0.1796	** 0.03 **	0.83	0.33
SLC38A2	0.4988 ± 0.0956	0.8955 ± 0.2930	0.5765 ± 0.1260	0.6828 ± 0.1806	0.27	0.48	0.57
** Cotyledon **							
SLC2A8	1.8630 ± 0.6801	2.3565 ± 0.6405	1.2701 ± 0.2903	2.3602 ± 0.4184	** 0.03 **	0.95	0.66
SLC43A2	0.9689 ± 0.2186	0.8167 ± 0.1563	0.7372 ± 0.1087	0.7485 ± 0.1581	0.63	0.78	0.74
SLC7A5	1.3560 ± 0.1220	2.0552 ± 0.5263	2.0696 ± 0.6891	2.2455 ± 0.3947	0.50	0.35	0.82
SLC7A8	1.0845 ± 0.2258	1.1113 ± 0.3165	1.7406 ± 0.3958	1.4869 ± 0.2127	0.57	** 0.02 **	0.35
SLC38A10	1.2280 ± 0.1337	1.4432 ± 0.1348	1.3228 ± 0.1407	1.6269 ± 0.1280	0.09	0.31	0.74
SLC38A2	0.7596 ± 0.1300	1.2885 ± 0.4477	1.0832 ± 0.3403	1.1360 ± 0.1771	0.14	0.45	0.96

Data are presented as means ± standard error. Bolded *p* values are significant at *p* ≤ 0.05. When there is a significant interaction, the *p* values for main effects of nutrition and treatment are not shown and replaced with a dash (-). Concentrations are expressed in µmol/L. Experimental units: ADQ-CON, n = 7; RES-CON, n = 7; ADQ-MEL, n = 7; RES-MEL n = 8. Abbreviations: NUT, nutrition; TRT, treatment; NUT*TRT, nutrition by treatment interaction; ADQ-CON, adequately fed control; RES-CON, restricted fed control; ADQ-MEL, adequately fed melatonin supplemented; RES-MEL, restricted fed melatonin supplemented.

**Table 5 metabolites-12-01208-t005:** Transcript abundance of nutrient transporters in caruncle and cotyledon tissue in the summer replicate.

Summer Trial					*p* Value		
Caruncle	ADQ-CON	RES-CON	ADQ-MEL	RES-MEL	NUT	TRT	NUT*TRT
SLC2A8	0.8645 ± 0.2889	0.9337 ± 0.2017	0.9708 ± 0.1910	0.7601 ± 0.1198	0.35	0.46	0.07
SLC43A2	0.8030 ± 0.1411	0.8314 ± 0.1520	1.2378 ± 0.4489	0.7084 ± 0.0592	0.78	0.55	0.98
SLC7A5	2.1421 ± 1.2201	1.7401 ± 0.4498	1.6200 ± 0.2052	1.1358 ± 0.2489	0.77	0.80	0.13
SLC7A8	0.8551 ± 0.2062	1.2899 ± 0.2353	1.0087 ± 0.2291	0.8653 ± 0.1672	0.54	0.52	0.18
SLC38A10	0.8291 ± 0.1746 ^b^	1.3427 ± 0.1732 ^a^	2.3497 ± 1.2254 ^ab^	0.9615 ± 0.1369 ^ab^	-	-	** 0.04 **
SLC38A2	0.3514 ± 0.0560 ^b^	0.7786 ± 0.132 ^a^	0.5043 ± 0.0877 ^a^	0.4226 ± 0.0812 ^b^	-	-	** <0.01 **
** Cotyledon **							
SLC2A8	1.8430 ± 0.1905	1.8302 ± 0.2693	1.6650 ± 0.1620	1.3944 ± 0.2035	0.29	0.22	0.49
SLC43A2	3.3572 ± 0.9071	5.4449 ± 1.2810	3.5628 ± 1.2810	2.4225 ± 1.1894	0.71	0.21	0.16
SLC7A5	1.9922 ± 0.9210	1.4450 ± 0.2738	0.7286 ± 0.1116	0.9518 ± 0.2310	0.27	0.09	0.63
SLC7A8	0.6057 ± 0.0706	0.5407 ± 0.0849	1.1313 ± 0.3202	0.8001 ± 0.2779	0.67	0.71	0.60
SLC38A10	0.8802 ± 0.2273	0.9702 ± 0.1828	1.7643 ± 0.4429	0.9625 ± 0.1304	0.55	0.17	0.12
SLC38A2	1.2958 ± 0.3566	1.3249 ± 0.4019	1.5967 ± 0.6684	1.9284 ± 1.2245	0.85	0.52	0.90

Data are presented as means ± standard error. Bolded *p* values are significant at *p* ≤ 0.05. When there is a significant interaction, the *p* values for main effects of nutrition and treatment are not shown and replaced with a dash (-). Superscripts (a, b) denotes differences at *p* ≤ 0.05. Concentrations are expressed in µmol/L. Experimental units: ADQ-CON, n = 6 RES-CON, n = 6; ADQ-MEL, n = 6; RES-MEL n = 7. Abbreviations: NUT, nutrition; TRT, treatment; NUT*TRT, nutrition by treatment interaction; ADQ-CON, adequately fed control; RES-CON, restricted fed control; ADQ-MEL, adequately fed melatonin supplemented; RES-MEL, restricted fed melatonin supplemented.

## Data Availability

Data is available in this article and within Appendix A.

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
