# Peer review of "Melatonin Supplementation Alters Maternal and Fetal Amino Acid Concentrations and Placental Nutrient Transporters in a Nutrient Restriction Bovine Model"

_metabolites, 2022, doi:10.3390/metabo12121208_

Round 1
Reviewer 1 Report
The current manuscript discusses the impact of restricted nutrition and melatonin treatment during different seasons on maternal and fetal amino acid concentrations and placental nutrient transporters. The authors concluded that melatonin subverted the effects of nutrient restriction on System A, N, Anion, Br, Bo, and total as well as essential maternal amino acids in the fall season as the results were comparable to ADQ-CON. Likewise, In summer, melatonin alleviated the effects of nutrient restriction on caruncle transporters (SLC38A10 and SLC38A2) and in Systems Br and Bo maternal amino acids. Authors concluded that melatonin if administered prior to parturition in nutrient-restricted cows could alleviate the negative impact of nutrient restriction in a seasonal-dependent manner.
Overall, the manuscript is well-written and designed to test the hypothesis. There are a few observations that must be addressed:
1. Authors must clearly describe the various parameters: System A, N Br, and BO in the materials and methods section; how they are relevant to the study.
2. Why did they not measure the blood flow to demonstrate that their model does have reduced blood flow due to season in nutritional-restricted model cows? Please discuss this limitation in the discussion
3. Also emphasize in materials and methods the steps taken to validate the nutrition restriction model. Please indicate various metabolic profiles of the glucose, BHBA, and other parameters before testing the amino acid profiles and transporter transcripts for each group.
Author Response
Reviewer 1:
- Since there are not individual placental nutrient transporters for each amino acid, we chose to evaluate amino acids by transport types. This may allow future directions to identify specific transporter dysfunction, or importance vs lack thereof for different types of amino acids in placental transport and fetal growth and development. Addressed in lines 145-150
- Uterine blood flow was measured on day 220 and published in a separate article by Contreras et. al, 2021. We do recognize the limitation of not collecting uterine blood flow on day 240 (when fluids and placentomes were collected for amino acid and nutrient transporter analysis). We also recognize a limitation of not collection umbilical blood flow to measure nutrient flux across the placenta. These are addressed in lines 352-355.
- Nutrient restriction model is validated from our previous work and cited in lines 106-108. Maternal nutrient restriction decreased pregnant heifer body weight and caused intrauterine growth restriction. As for other metabolic profiles, another researcher in our lab is preparing a manuscript covering fatty acids from this study. We first targeted amino acids due to profound differences observed in branched chain amino acids in a similar study from our lab using sheep (Lemley et al., 2013).
Reviewer 2 Report
Swanson et al. have studeied the melatonin supplementation on maternal and fetal amino acid concentrations and placental nutrient transporters in a nutrient restriction bovine model. It is an interesting topic. The experimental was well-desginsed. The paper was well-written. The following changes could improve the quality of the paper.
1. please remove ‘.’ from the title.
2. Line 71, remove the addtional sapce in 'rats [13,16]'.
3. Line 89, 194 etc., checking the writting of '±' throughout the paper and unified them.
4. Line 183, etc., checking the writting of 'P' throughout the paper. It should be written in italic.
5. Line 230, please unified the effective numbers of the P value. Too much effective unmber after the decimal point is not meaningful. The same for other values.
6. Table 2, checking the presentation of the Table. It seems not good presented.
7. Please add the replicates (n=?), values are means ± SD or SE?, P value setuo for the significant changes, full name for the abbreviations etc., in the footnotes of all the tables.
8. Checking the referens and make sure following the journal style.
Author Response
Reviewer 2:
- Addressed
- Addressed
- Addressed
- Addressed
- Addressed – changed to 2 decimal points throughout manuscript, including tables
- Addressed in comment 7.
- Addressed – added all of the requested information to footnotes.
- Addressed – added missing DOIs and middle initials